# OpenReview forum: "SmoothLLM: Defending Large Language Models Against Jailbreaking Attacks"
_TMLR — Accepted by TMLR_

### Review · Reviewer_qxkL · 2025-02-04

**Summary Of Contributions:**

The paper introduces a new defense approach, called SmoothLLM, for mitigating jailbreaking attacks on large language models (LLMs). Jailbreaking attacks manipulate input prompts to bypass safety measures, causing LLMs to generate harmful or restricted content. SmoothLLM addresses this vulnerability by perturbing multiple copies of an input prompt at the character level and aggregating the model’s responses to detect adversarial inputs. The paper demonstrates that SmoothLLM can reduce attack success rates across multiple jailbreak methods and achieve state-of-the-art robustness. The approach is compatible with both black-box and white-box LLMs, does not require retraining, and maintains efficiency with a small trade-off in nominal performance. The paper also evaluates adaptive attacks and finds that SmoothLLM remains resilient while maintaining its effectiveness against jailbreaking attacks.

**Audience:**

Yes

**Claims And Evidence:**

Yes

**Requested Changes:**

1. Add discussion and/or experiments on how to mitigate performance degradation while preserving defense effectiveness.
2. Evaluate on a stronger adaptive attack against SmoothLLM.
3. Discuss potential latency issues in real-time applications and strategies to minimize computational costs.

**Strengths And Weaknesses:**

### Strengths
1. It is an interesting adaptation of randomized smoothing from the computer vision domain to LLMs for defending against jailbreaking attacks.
2. The proposed approach is evaluated against multiple state-of-the-art attacks and demonstrates strong defensive performance across various LLMs.
3. The defense is efficient and scalable, capable of mitigating jailbreaks with a small number of additional queries.
4. The paper provides a theoretical analysis of SmoothLLM’s robustness.


### Weaknesses
1. SmoothLLM is effective against adversarial suffixes composed of random characters, as these jailbreak prompts are less robust to character-level perturbation. However, for semantic jailbreaks such as PAIR, it fails to sufficiently mitigate the attack. For instance, the attack success rate remains 47% on Vicuna and 25% on GPT-4.
2. There is a trade-off between robustness and nominal model performance, particularly at higher perturbation levels. However, even with the lowest perturbation percentage, there is still a non-trivial performance drop, as shown in Figure 9. Further exploration of how to mitigate performance degradation while preserving defense effectiveness would be beneficial.
3. While the authors test adaptive attacks on SmoothLLM, they conclude that these attacks do not significantly improve over non-adaptive attacks. However, the adaptive attack applies GCG on a SmoothLLM surrogate and then transfers to the actual protected model. First, as shown in Table 1, GCG is less effective against semantic attacks like PAIR. Therefore, the conclusion is not accurate. The adaptive attack should build on top of the strongest existing attack. Second, to evade a defense that is not differentiable, one can leverage the Monte Carlo method. For example, the attack can use PAIR to generate a jailbreak prompt while applying multiple rounds of SmoothLLM. If the attack fails, it can continue running PAIR on top of the SmoothLLM-perturbed prompts. This process can iterate multiple times until it succeeds.
4. Similar to other randomized smoothing-based methods, SmoothLLM introduces an increased number of queries. The paper could discuss potential latency issues in real-time applications and strategies to minimize computational costs.

---

> ### Author Response · Authors · 2025-02-18
> **Review response**
>
> Thanks so much for your review.  Below, we address the three points in the "requested changes" section of your review."
>
> ---
>
> > Add discussion and/or experiments on how to mitigate performance degradation while preserving defense effectiveness.
>
> Thanks for this comment. In Appendix E, we have an explanation for why performance degrades at high perturbation levels. Additionally, we added the following discussion to Appendix B.13:
>
> ```
> A primary factor behind SmoothLLM's nominal performance degradation that results is the fact that random character-level perturbations degrade the semantic content of input prompts. This problem was also observed in randomized smoothing, and various solutions in that literature were proposed. A notable solution was to prepend a denoising/smoothing function to a ML model, yielding the paradigm of *denoised smoothing*. Recent work has shown that when applied to SmoothLLM, the paradigm of denoised smoothing results in significantly less degraded nominal performance [A].  In particular, the Pareto curves swept out in Figure 2 of [A] shows that SmoothLLM can be implemented with semantic, rather than character-level, perturbations, yielding significantly better nominal performance.
> ```
>
> [A] [Defending Large Language MOdels against Jailbreak Attacks via Semantic Smoothing](https://arxiv.org/pdf/2402.16192)
>
> ---
>
> > "First, as shown in Table 1, GCG is less effective against semantic attacks like PAIR. Therefore, the conclusion is not accurate.
>
> We're a bit confused. GCG and PAIR are both attacks, so GCG cannot be "effective against" PAIR.  They are both algorithms designed to jailbreak a target model. Can you elaborate a bit more here?
>
>
> > Second, to evade a defense that is not differentiable, one can leverage the Monte Carlo method. For example, the attack can use PAIR to generate a jailbreak prompt while applying multiple rounds of SmoothLLM. If the attack fails, it can continue running PAIR on top of the SmoothLLM-perturbed prompts. This process can iterate multiple times until it succeeds."
>
> We're not sure that we follow this. Could you perhaps explain in more detail? We're not clear on what you mean by "applying multiple rounds" of SmoothLLM.
>
> ---
>
> > "The paper could discuss potential latency issues in real-time applications and strategies to minimize computational costs."
>
> Thanks for your comment. In Table 8 in Appendix B, we provide an analysis of the latency of running SmoothLLM. This table shows the increase in latency incurred by increasing the number of perturbed copies.
>
> There are several strategies that could be considered when deploying SmoothLLM in a real-world application. For example, representations could be cached at inference time (using, e.g., KV cache); this could enable higher throughput, especially for lower perturbation percentages, which may be mapped to relatively similar represetation spaces.  Furthermore, one could batch process the copies from SmoothLLM (using, e.g., the VLLM package), which would decrease latency via parallelization.  Generally, $N$ is small enough to enable all perturbed copies to fit in a single batch.  Finally, one could also perturb one by one until a potential jailbreak is found. This would potentially require fewer calls to the LLM.

---

> > ### Comment · Reviewer_qxkL · 2025-02-24
> > **Thanks for the response**
> >
> > Thank you, authors, for your response and for addressing the comments.
> >
> > There was a typo in the sentence: "GCG is less effective against semantic attacks like PAIR." It should be "less effective than." The point is that you may want to try PAIR to construct an adaptive attack since it is more effective in the non-adaptive setting.
> >
> > Regarding "applying multiple rounds," you first generate a jailbreak prompt and attach it to the toxic query. Then, you apply SmoothLLM to this jailbreak input. Obviously, this input will be rejected. Since it is perturbed by SmoothLLM, you can run GCG or PAIR to update the jailbreak prompt on top of this perturbed input. You repeat this process multiple times to see if it can eventually bypass SmoothLLM.

---

> > > ### Author Response · Authors · 2025-02-24
> > > **Quick response**
> > >
> > > Thanks for your response! A couple of quick comments on our end:
> > >
> > > ---
> > >
> > > > "The point is that you may want to try PAIR to construct an adaptive attack since it is more effective in the non-adaptive setting."
> > >
> > > Thanks for clarifying, we agree that this makes sense.  We would be happy to run this, but before doing so, we note that several follow-up works have run this exact experiment:
> > >
> > > * Table 3 in [Defending Large Language Models against Jailbreak Attacks via Semantic Smoothing](https://arxiv.org/pdf/2402.16192)
> > > * Section B.1, Algorithm 1, and Figure 15 (the metric here is *loss* rather than ASR) in [Improved Few-Shot Jailbreaking Can Circumvent Aligned Language Models and Their Defenses](https://arxiv.org/pdf/2406.01288)
> > >
> > > We're happy to rerun, but we think that these independent reproductions of these results may be just as, if not more, convincing.  Please let us know what you think.  Another option: We can reference these works in our experiments rather than reproducing their plots/tables.
> > >
> > > ---
> > >
> > > > "Regarding "applying multiple rounds," you first generate a jailbreak prompt and attach it to the toxic query. Then, you apply SmoothLLM to this jailbreak input. Obviously, this input will be rejected. Since it is perturbed by SmoothLLM, you can run GCG or PAIR to update the jailbreak prompt on top of this perturbed input. You repeat this process multiple times to see if it can eventually bypass SmoothLLM."
> > >
> > > Thanks for clarifying. We strongly suspect that if the attacker is allowed to attack the defended prompt, then the attack will likely be successful in most cases.  This will be particularly true for PAIR, since PAIR is allowed to rephrase the entire prompt.  We note that this kind of attack, while interesting, doesn't fit within the threat model outlined in Figure 3. But please do let us know if you feel strongly that this attack should be included.
> > >
> > > Another point of view: Running GCG takes around ~90-120 minutes per suffix on an A100, and so if we were to iterate, it will take hours, which (in our view) makes this attack relatively unreasonable beyond providing an academic proof of concept. PAIR would of course be faster, but as mentioned before, SmoothLLM is likely to be ineffective for any threat model in which the attacker is allowed to play last.

---

> > > > ### Comment · Reviewer_qxkL · 2025-02-25
> > > > **Thanks for the quick response**
> > > >
> > > > If existing work has already reported similar results, please reference these works in the paper so readers have a comprehensive understanding of the landscape.
> > > >
> > > > If you believe this adaptive attack is likely to evade SmoothLLM, it would be more informative for readers if you include the discussion in the paper.

---

> > > > > ### Author Response · Authors · 2025-02-25
> > > > > **Quick response II**
> > > > >
> > > > > Thanks again for your response.
> > > > >
> > > > > ---
> > > > >
> > > > > > "If existing work has already reported similar results, please reference these works in the paper so readers have a comprehensive understanding of the landscape."
> > > > >
> > > > > We added this paragraph to Section 4.1.
> > > > >
> > > > > ```
> > > > > Since the public release of this paper, several follow-up works have studied the robustness of \textsc{SmoothLLM} against different adaptive attacks. For instance,~\citep{ji2024defending} report the ASRs of \textsc{SmoothLLM} under adaptive \textsc{PAIR} and \textsc{AutoDAN} attacks, finding that \textsc{SmoothLLM} is more susceptible to these adaptive attacks than it is to the \textsc{GCG} adaptive attacks derived in this paper.  Similarly,~\citep{pang2025improved} derive a distinct differentiable GCG-like scheme for adaptively attacking \textsc{SmoothLLM} (see Algorithm 2 in~\cite{pang2025improved}, showing similar trends in the loss space to Figure~\ref{fig:smoothing-ASR}.
> > > > > ```
> > > > >
> > > > > ---
> > > > >
> > > > > > "If you believe this adaptive attack is likely to evade SmoothLLM, it would be more informative for readers if you include the discussion in the paper."
> > > > >
> > > > > We added this note to Appendix C.3.
> > > > >
> > > > > ```
> > > > > Adaptive attack that are likely to bypass \textsc{SmoothLLM} are those that allow the attacker to play after a forward pass through \textsc{SmoothLLM}. While this does not fit within the threat model described in Figure~\ref{fig:defense-schematic}, it's worth noting that any attack that can modify the prompt produced by the majority voting step in an unconstrained way will tend to succeed. We are not currently aware of any such attacks, but if one was given white-box access to a \textsc{SmoothLLM} implementation, such an attack would be theoretically possible.
> > > > > ```

---

### Review · Reviewer_eqZ8 · 2025-02-05

**Summary Of Contributions:**

SmoothLLM has become the most popular defense baseline for jailbreak attack. As an initial attempt to solve the jailbreak attack, this paper has offered a lot insight for the subsequent research on jailbreak attack. It deserves and definitely should be recognized as an important contribution to the community.

**Audience:**

Yes

**Claims And Evidence:**

Yes

**Requested Changes:**

1. Include a discussion to RA-LLM.
2. Include an experiment for test-time computation tradeoff.
3. Include discussion on randomnized smoothing,
4. Include more discussion on design motivation.
5. Include a discussion to new type of jailbreak attack.

**Strengths And Weaknesses:**

* Strengths:
1. The proposed method is simple and intuitive enough. The authors have make a lot of efforts in validate the effectiveness of their methods.
2. Experiment is very comprehensive.


* Weakness:
1. An important relevant work is missing. The design of SmoothLLM is similar to RA-LLM[1]. [1] is firat available at Sep 18, 2023, which is earlier than smoothLLM in any way. It is unfair for the authors of RA-LLM, as this paper even without including a reference to them. I suggest the authors to include a speficic paragraph to discuss the similarity and difference between smoothLLM and RA-LLM.

[1] Cao, Bochuan, et al. "Defending against alignment-breaking attacks via robustly aligned llm." arXiv preprint arXiv:2309.14348 (2023).

2. Test-time computation overhead. This weakness is obvious and has been discussed by many paper after smoothLLM. I suggest the authors to include a table to show the tradeoff between computation overhead and defense performance (i.e., show by adjusting hyper-parameter to tradeoff between computation and defense performance). I don't think extra computation is a big issue for a research work, but the authors should provide information on how the tradeoff looks like.

3. Relationship with randomnized smoothing is not explicitly discussed. SmoothLLM is inspired by randomnized smoothing[2]. I suggest the author to add a section in appendix to discuss how jailbreak attack is different and similar to adversarial attack  and how smoothLLM is different and similar to randomnized smoothing.

[2] Certified Adversarial Robustness via Randomized Smoothing

4. A statement need more discussion. Page 6, it is written that "The motivation for this step is that while one perturbed prompt may not mitigate an attack,as evinced by Figure 4, on average, perturbed prompts tend to nullify jailbreaks. That is, by perturbing
multiple copies of each prompt, we are likely to flip characters in the adversarial portion of the prompt." What if the length of the adversarial suffix is pretty short? In this case the chance of perturbing those suffix will be smaller and therefore smoothLLM will be less successful? Moreover, why  perturbing multiple copies of each prompt help? Perturbing multiple copies does not increase the expectation that the suffix will be perturbed, but it just increase the trial number. The expectation of  perturbing the suffix should be the same.  If the expectations are the the same, why do you mean by "we are likely to flip characters in the adversarial ..."


5. Discussion towards a new type of jailbreak attack. Recently, there a new type of jailbreak attack called fine-tuning attack [3]. I suggest the authors to add a section in the appendix for discussion. I didn't expect smoothLLM to be able to handle the fine-tuning jailbreak, and it is unfair to request the authors for extra experiments on them, but I do expect a few comments and insights on how to apply smoothLLM on fine-tuning attack.  I also suggest the authors to discuss about some recent defenses in the same section, e.g., [4-10].

[3] Fine-tuning aligned language models compromises safety, even when users do not intend to!"

[4] Vaccine: Perturbation-aware alignment for large language model aginst harmful fine-tuning attack

[5] Representation noising effectively prevents harmful fine-tuning on LLMs

[6] Keeping llms aligned after fine-tuning: The crucial role of prompt templates

[7] Mitigating fine-tuning jailbreak attack with backdoor enhanced alignment

[8] Lisa: Lazy safety alignment for large language models against harmful fine-tuning attack

[9] Safe lora: the silver lining of reducing safety risks when fine-tuning large language models

[10] Navigating the safety landscape: Measuring risks in finetuning large language models

---

> ### Author Response · Authors · 2025-02-18
> **Review response**
>
> Thanks so much for your review.  Below we answer your questions and provide evidence that we hope resolves your concerns.
>
> ---
>
> > "An important relevant work is missing. The design of SmoothLLM is similar to RA-LLM[1]. . ."
>
> Thanks for pointing this out.  To clarify, these papers appeared ***nearly* concurrently** on arXiv: RA-LLM first appeared on **Sept. 18, 2023**, whereas SmoothLLM first appeard **17 days later** on **Oct. 5, 2023**. We agree that RA-LLM should be discussed. We added the following discussion in Section 2.3 of our submission:
>
> ```
> Also related is the near-concurrent work of (Cao et al, 2023), which introduces an algorithm called RA-LLM that applies randomly generated masks to input prompts. Similarly to SmoothLLM, RA-LLM also adopts a modify-then-aggregate algorithm, although SmoothLLM uses different perturbation functions: inserts, swaps, and patches.  Moreover, we investigate how nominal performance and latency vary with various hyperparameters; (Cao et al, 2023) leave such experiments to future work.
> ```
>
> Let us know if this resolves your concern.
>
> ---
>
> > "Test-time computation overhead. This weakness is obvious and has been discussed by many paper after smoothLLM. I suggest the authors to include a table to show the tradeoff between computation overhead and defense performance (i.e., show by adjusting hyper-parameter to tradeoff between computation and defense performance)."
>
> Thanks for your comment. This evidence is provided in Table 8 in Appendix B of our submission.
>
> ---
>
> > "Relationship with randomnized smoothing is not explicitly discussed. SmoothLLM is inspired by randomnized smoothing[2]. I suggest the author to add a section in appendix to discuss how jailbreak attack is different and similar to adversarial attack and how smoothLLM is different and similar to randomnized smoothing."
>
> Thanks for your comment. We have added a new appendix—**Appendix F**—which discusses the similarities and differences between classical randomized smoothing and SmoothLLM (see Appendix F.2).
>
> ---
>
> > "A statement need more discussion. . . What if the length of the adversarial suffix is pretty short? In this case the chance of perturbing those suffix will be smaller and therefore smoothLLM will be less successful? Moreover, why perturbing multiple copies of each prompt help? Perturbing multiple copies does not increase the expectation that the suffix will be perturbed, but it just increase the trial number. The expectation of perturbing the suffix should be the same. If the expectations are the the same, why do you mean by "we are likely to flip characters in the adversarial ...""
>
> Thanks for you questions. We answer them sequentially.
>
> 1. **Suffix length.** If the length of the suffix is short (and assuming that the length of the nominal prompt stays fixed), it's true that the probability of perturbing the suffix goes down.  While this may reduce SmoothLLM's effectiveness, it's also true that shorter suffixes may be less likely to result in jailbreaks, so it's unclear to us what would happen in this case. We think this is a fruitful direction for future work, and, notably, so do the authors of [1], who write that "we have not assessed how our method performs when faced with very extreme cases, such as particularly long or short adversarial prompts."
> 2. **Multiple copies.** Perturbing multiple copies has several effects. Firstly, if the threshold is set to be reltatively large (i.e., $\gamma = (N-1)/N$), perturbing multiple copies allows us to block jailbreaks even if they are detected in a single one of the copies. We provide evidence for this in Table 7 in Appendix B.  Secondly, we note that empirically, SmoothLLM's performance improves considerably with increasing $N$; see Figure 7 for evidence.  We attribute this result to the fact that using multiple copies reduces the variance of the Monte Carlo estimate in (4).
>
> ---
>
> > "Discussion towards a new type of jailbreak attack. Recently, there a new type of jailbreak attack called fine-tuning attack [3]. I suggest the authors to add a section in the appendix for discussion. . . I also suggest the authors to discuss about some recent defenses in the same section, e.g., [4-10]."
>
> Thanks for your comment. We agree that these jailbreaks should be discussed. We have added a discussion of fine-tuning attacks (including references [4-10]) in Appendix F.4.

---

> > ### Comment · Reviewer_eqZ8 · 2025-02-24
> > **Thanks for the rebuttal**
> >
> > Hi authors,
> >
> > Thanks for the rebuttal. It seems that you forgot to update the discussion of (Cao et al,2023) in the new revision  in Section 2.3. Please do update it if possible.
> >
> > I believe the smoothLLM paper has already been well polished  before submission to TMLR, and therefore I don't have much opinion on its acceptance. Indeed, many of the new defenses/attacks idea for LLM can be traced back to those for DNN. However, what I generally believe is that as long as the defense/attack is usable and effective, then it is a valuable work. We should not nitpick too much on the so-called "novelty".
> >
> > Because most of the idea is inspired by the vanilla idea, it is very possible that two con-current papers  work on the same idea. However, it is important to respect the con-current work by adding reference and discussion to them as this is important for the well-being of community. Simply pretending not seeing the con-current work is definitely not a good practice. I appreciate the authors willingness to  recognize the con-current work in the revision.

---

> > > ### Author Response · Authors · 2025-02-24
> > > **Quick response**
> > >
> > > Thanks for your response! We just added that paragraph in our rebuttal that discusses (Cao et al., 2023). We completely agree that it's worthwhile and necessary to discuss concurrent work, and there was no effort on our part to knowingly ignore the work of (Cao et al., 2023) in our original submission.

---

### Review · Reviewer_fPHo · 2025-02-18

**Summary Of Contributions:**

The paper investigates jailbreaks, so far, a commonly known issues of the model. they propose SmoothLLM, which perturbs input prompts at the character level, runs multiple copies, and aggregates results to detect adversarial inputs. Testing shows it significantly reduces attack success for several jailbreak methods, including newer ones like RandomSearch and AmpleGCG. It works without retraining, has a small trade-off in regular performance, and is compatible with both black-box and white-box LLMs, making it a practical defense against jailbreak attacks.

**Audience:**

Yes

**Broader Impact Concerns:**

none.

**Claims And Evidence:**

No

**Requested Changes:**

please respond to the 3rd and 4th weakness above.

**Strengths And Weaknesses:**

strengths:

+ the idea is more like a LLM-adapted version of randomized smoothing, which has potentially significant values as randomized smoothing has introduced.

+ the paper has written in a very smooth manner, very easy to follow.

weakness:

- while the paper has innovation values (as it might be considered as the first one to bring the idea of randomized smoothing into jailbreak), this is more like a position-wise innovation. The actual value of technical innovation seems fairly limited other than bringing the ideas into LLM world. Even the authors themselves seem to have issues in summarizing the contributions of their own paper (bullet points on Page 2 seem vague).

- The "Desiderata" listed stands fine for the scope of this paper, but the authors seem to argue that these desiderata should hold for jailbreak study, in this case, this claim will need a lot more justifications and support.

- The "efficiency" part is also not so justified well, as the main argument is about the cost of smoothLLM vs. the cost of undefended LLM, and cost of smoothLLM vs. the cost of generating an attack. However, I will argue the point that really matters is the cost of smoothLLM vs. the cost of other defense methods. For example, if smoothLLM is the LLM version of randomized smoothing, [1] is the LLM version of adversarial training. It will be much more interesting to compare these two.

- Another question I have is the regarding the claim "The suffixes generated by adversarial suffix jailbreaks are fragile to character-level perturbations" (Section 3.1) this will be true for GCG style prompts, but I would doubt whether this claims will hold for DAN style prompts. The authors made a pretty strong arguments on this, and it will be necessary to offer such justifications over all other style prompts.
     - it will be better to directly show the empirical results over different families of jailbreak prompts.
     - the authors might respond in a way they are only focusing on the GCG style prompts, however, the overall writing of this paper indicate they are studying overall jailbreaks except some mentioning of "adversarial prompting". For example, the above claim is very strong and didn't specify the scope, there are many unscoped remarks and arguments like this, making the claims of this paper questionable. Therefore, I would recommend the authors to show empirical results over different categories of the jailbreaks to fit the claims, or maybe fix these claims if the authors prefer.


[1] Robust prompt optimization for defending language models against jailbreaking attacks

---

> ### Author Response · Authors · 2025-02-18
> **Review response**
>
> Thanks so much for your review.  Below, as requested, we respond to your 3rd and 4th bullet points.
>
> ---
>
> > "The "efficiency" part is also not so justified well. . . if smoothLLM is the LLM version of randomized smoothing, [1] is the LLM version of adversarial training. It will be much more interesting to compare these two."
>
> Thanks for your question. Here's the requested analysis:
>
> To obtain an RPO suffix, the author's implementation of Algorithm 1 in [1] runs for $T=500$ steps, where each step requires the computation of the gradient of the loss. To compute this loss, the algorithm runs one of GCG, PAIR, or a manual jailbreak as a subroutine in each loop iteration.
> * Running GCG requires (500 steps per prompt) $\times$ ($m=25$ prompts); see Section 4.1 in [1].  Factoring in the additional "safe loss" gradient computation, obtaining an RPO suffix requires $500\times (500 \times 25 +1) = 6250500$ gradient computations/forward passes through the model.  [1] does not provide a timinganalysis, but given that RPO is the defense analog of GCG, we can extrapolate the results for Llama2-7b in Section 4.3 of our submission to estimate that obtaining such a single suffix may take on the order of *days* for the GCG variant.
> * Running PAIR or JBC would require considerably less compute. PAIR requires roughly 60 calls split between three LLMs---the target, the attacker, and the judge---per iteration, and JBC would have minimal (if any) overhead.  For PAIR, if we roughly estimate that finding a jailbreak takes 1 min (wall clock), then running RPO for 500 steps would take around 600 minutes. JBC likely has the same running time as GCG, and therefore takes 1.5-2 hours on an A100.
>
> In contrast, SmoothLLM can defend against the same attacks using ~3 seconds of additional computation. Relative to GCG, as reported in Table 1 of [1], in the worst case, SmoothLLM matches the performance of RPO in several cases, and is only ever outperformed by RPO by at most 3 percentage points.  So more or less, **SmoothLLM matches RPO's performance on GCG (as reported in [1]) using several orders of magnitude less compute.** We would be happy to include this analysis in the paper if you find it helpful.
>
> ---
>
> > "Another question I have is the regarding the claim "The suffixes generated by adversarial suffix jailbreaks are fragile to character-level perturbations" (Section 3.1) this will be true for GCG style prompts, but I would doubt whether this claims will hold for DAN style prompts. The authors made a pretty strong arguments on this, and it will be necessary to offer such justifications over all other style prompts."
>
> Thanks for your question. We never claimed that this finding holds for DAN-style prompts. Our claim (as you quoted) is that **suffix-based** jailbreaks are fragile to character-level perturbations; DAN is **not** a suffix-based jailbreak. We provide evidence to support our claim regarding **suffix-based** jailbreaks in Figure 4. If there are any claims that you think we should tone down, we are happy to do so.
>
> ---
>
> > "it will be better to directly show the empirical results over different families of jailbreak prompts."
>
> Thanks for you comment. We provide evidence for different jailbreak families in Figure 1.  SmoothLLM is most effective against suffix-based jailbreaks; see GCG (top), RandomSearch (bottom middle), and AmpleGCG (bottom right). It is effective, but to a lesser extent, against PAIR (bottom right), which is **not** suffix-based.
>
> ---
>
> > "the authors might respond in a way they are only focusing on the GCG style prompts, however, the overall writing of this paper indicate they are studying overall jailbreaks except some mentioning of "adversarial prompting". For example, the above claim is very strong and didn't specify the scope, there are many unscoped remarks and arguments like this, making the claims of this paper questionable. Therefore, I would recommend the authors to show empirical results over different categories of the jailbreaks to fit the claims, or maybe fix these claims if the authors prefer."
>
> Thanks for your comment. Again, the above claim (that you quoted) specifically pertains to **suffix-based** jailbreaks.  And as mentioned above, we show various results against PAIR, which is **not** a suffix-based jailbreak.  Based on your feedback, we have updated the following claims:
>
> 1. *Introduction:* "Based on this desiderata, we next introduce SmoothLLM, the first algorithm designed to mitigate jailbreaking attacks" --> "Based on this desiderata, we next introduce SmoothLLM, a jailbreaking defense algorithm that is particularly effective against suffix-based attacks."
> 2. *Section 3:* "Given the need to design defenses against jailbreaking attacks, we propose SmoothLLM." --> "Given the need to design defenses against jailbreaking attacks, we propose SmoothLLM, which is particularly well-suited to defending against suffix-based jailbreaks."
>
> Let us know if this resolves your concern.

---

### Decision · Action_Editor_q6zc · 2025-04-06

**Recommendation:** Accept with minor revision

**Comment:**

In this 1st-round rebuttal, the reviewers confirmed the novelty and technical contribution of SmoothLLM, but also proposed a lot of suggestions like some non-clarified details, better explanation or re-organization and more experiments as well as tuning the scope in writing. After rebuttal, most concerns of the reviewers have been addressed, but some remain, which leads to a diverging decision (both rejection and acceptance). Through a converging discussion between AE and reviewers, we reach to the consensus that we can give some space in this community given the idea and technical contribution of this submission,  but with the requirement in the following points,

1. comprehensively refine the scope including the title and other description in the paragraphs for precise claims.

2. do an experiment to compare SmoothLLM and RPO, instead of direct number comparison in two papers given the different settings.

Based on the discussion, AE tends to gives the recommendation of "Acceptance with minor revision". Hope the authors carefully follow the suggestions, which is summarized by AE from the reviewers' remaining concerns.

Best,

AE

**Audience:**

The researchers or readers that are interested in the attack and defense research of LLMs, which fits the range of TMLR's audience.

**Claims And Evidence:**

This submission focuses on defending the jailbreak of LLMs and proposed a SmoothLLM method, which randomly perturbs multiple copies of input prompt and then aggregates together to improve the robustness against GCG, PAIR etc. A range of experiments show the effectiveness of the proposed method, compared with the current state-of-the-art methods.